# F-Wave Extraction from Single-Lead Electrocardiogram Signals with Atrial Fibrillation by Utilizing an Optimized Resonance-Based Signal Decomposition Method

**DOI:** 10.3390/e24060812

**Published:** 2022-06-10

**Authors:** Junjiang Zhu, Jintao Lv, Dongdong Kong

**Affiliations:** 1School of Mechanical and Electrical Engineering, China Jiliang University, Hangzhou 310018, China; zhujunjiang@cjlu.edu.cn (J.Z.); s20010811016@cjlu.edu.cn (J.L.); 2School of Mechatronic Engineering and Automation, Shanghai University, 99 Shangda Road, Baoshan District, Shanghai 200444, China

**Keywords:** F-wave extraction, atrial fibrillation, resonance-based signal decomposition, morphological component analysis, wavelet transform, genetic algorithm

## Abstract

(1) Background: A typical cardiac cycle consists of a P-wave, a QRS complex, and a T-wave, and these waves are perfectly shown in electrocardiogram signals (ECG). When atrial fibrillation (AF) occurs, P-waves disappear, and F-waves emerge. F-waves contain information on the cause of atrial fibrillation. Therefore it is essential to extract F-waves from the ECG signal. However, F-waves overlap the QRS complex and T-waves in both the time and frequency domain, causing this matter to be a difficult one. (2) Methods: This paper presents an optimized resonance-based signal decomposition method for detecting F-waves in single-lead ECG signals with atrial fibrillation (AF). It represents the ECG signal utilizing morphological component analysis as a linear combination of a finite number of components selected from the high-resonance and low-resonance dictionaries, respectively. The linear combination of components in the low-resonance dictionary reconstructs the oscillatory part (F-wave) of the ECG signal. In contrast, the linear combination of components in the high-resonance dictionary reconstructs the transient components part (QRST wave). The tunable Q-factor wavelet transform generates the high and low resonance dictionaries, with a high Q-factor producing a high resonance dictionary and a low Q-factor producing a low resonance dictionary. The different Q-factor settings affect the dictionaries’ characteristics, hence the F-wave extraction. A genetic algorithm was used to optimize the Q-factor selection to select the optimal Q-factor. (3) Results: The presented method helps reduce RMSE between the extracted and the simulated F-waves compared to average beat subtraction (ABS) and principal component analysis (PCA). According to the amplitude of the F-wave, RMSE is reduced by 0.24–0.32. Moreover, the dominant frequency of F-waves extracted by the presented method is clearer and more resistant to interference. The presented method outperforms the other two methods, ABS and PCA, in F-wave extraction from AF-ECG signals with the ventricular premature heartbeat. (4) Conclusion: The proposed method can potentially improve the accuracy of F-wave extraction for mobile ECG monitoring equipment, especially those with fewer leads.

## 1. Introduction

Atrial fibrillation (AF) is the most common persistent arrhythmia, with 33.5 million patients all over the world [1,2]. AF directly endangers the health of patients and also increases the risk of coronary heart disease, hypertension, and heart failure. Therefore, the diagnosis and monitoring of AF patients are of great significance. In clinical practice, a body surface electrocardiogram (ECG) is often used to monitor AF patients. Under normal circumstances, the ECG signal consists of P-wave generated by atrial activity and QRS-wave and T-wave caused by ventricular activity. When AF occurs, the ECG signal will be manifested as an RR-interval disorder and disappearance of the P-wave. Additionally, the serrated F-wave appears instead of the P-wave. By analyzing the frequency spectrum and amplitude of the F-wave, not only can we distinguish the type of AF but also judge whether AF relapses. However, how to extract F-waves from ECG becomes a difficult problem since F-waves overlap with QRST-wave in time and space [3].

With regards to F-wave extraction in multi-lead ECG signals, Stridh et al. [4,5] proposed a spatiotemporal QRST elimination method. This method utilizes spatial transformation to accurately align the ECG signals of different leads, then adds all the QRST items to get the average template, and finally subtracts the template from the original signal to get the AF-wave. Principal component analysis (PCA) [6,7] and independent component analysis (ICA) [8,9,10] determine a set of basic components according to the intrinsic relationship of ECG vectors in different leads and then use the amplitude or dominant frequency of AF-wave to select the components for a reconstruction of AF. These methods start from the spatial composition of ECG signal and are insensitive to QRST waveform change and noise. Petrenas [11] and Mateo [12] proposed a neural network extraction algorithm by adopting the lead ECG data far away from the atrium as the reference to train the QRST template. These methods adopt the correlation between leads as prior information. However, they cannot guide the extraction of AF-wave in single-lead ECG. With regards to AF-wave extraction from single-lead ECG signal, the template elimination method [13] is similar to the spatiotemporal QRST elimination method. They both adopt multiple heartbeat data to construct the QRST template and then realize AF-wave extraction by eliminating QRST-wave, which is simple and easy to operate. However, they are sensitive to QRST waveform changes. In recent years, with the development of wearable medical devices, researchers began to focus on how to use certain characteristics of AF signals themselves to improve the extraction effect of AF-wave.In reference [14], the heartbeat similarity of the single-lead ECG signal was used to segment the signal, which was then decomposed. Next, the AF-wave was extracted based on the non-stationary characteristics of AF. However, AF patients have great individual differences in ECG signals and are often accompanied by diseases such as ventricular premature contraction and indoor differential conduction, which may affect the effect of AF-wave extraction methods. The F-wave has the characteristics of the harmonic signal, while the QRST-wave has the characteristics of periodic shock. However, to the author’s knowledge, there is no relevant research on the method of extracting F-waves according to this law.

In the fields of feature extraction in EEG signals [15,16] and fault feature extraction in mechanical vibration signals [17,18], there are cases where different signal components are divided from the perspective of shape. Morphological component analysis (MCA) [19] is a compelling new method to solve the problem of signal and image feature separation [20]. MCA uses the differences between the different components of the signal to separate the signal. Tunable Q-factor wavelet transform (TQWT) [21] enables the tunable Q-factor wavelet to achieve optimal matching for signals with specific oscillation properties by selecting Q-values. After the TQWT processes the signal, it exhibits sparse properties in the wavelet subbands.

MCA decomposition relies on two dictionaries with different shape types, while tunable wavelets can be used to generate dictionaries with varying forms of oscillation. Considering the different changes of F-waves and QRST waves, this paper utilizes the resonance-based signal decomposition method to carry out F-wave extraction from the single-lead central electrical signal. In this paper, high-resonance and low-resonance dictionaries are obtained through TWQT. The linear combination of the components in the high-resonance dictionary reconstructs the oscillatory part of the ECG signal, and the linear combination of the components in the low-resonance dictionary reconstructs the transient components part. Meanwhile, MCA represents the signal as a linear combination of a finite number of components by sparse decomposition, picked from high-resonance and low-resonance dictionaries, respectively. However, the dictionaries produced under different Q-factors are different. Under the influence of the characteristic laws of F-waves and QRST waves, the effect of the high and low resonance dictionary of unselected Q-factors on the extraction effect of F-waves did not reach good expectations. To obtain better high-low resonance dictionary pairs, this paper uses a genetic algorithm to optimize the selection of the Q-factor.

## 2. Methods

### 2.1. Resonance-Based Signal Decomposition

This section provides a comprehensive introduction to the optimized resonance-based signal decomposition method, which will be utilized for F-wave extraction in Section 3.

The morphological component analysis represents the signal sparsely as a sum of several components. These components are selected from two dictionaries with different waveform morphological types. The components of the various dictionaries are used to reconstruct the components of the signal with varying waveform characteristics. The resonant sparse decomposition of the signal is a special type of morphological component analysis obtained using the tunable Q-factor wavelet transform for both dictionaries. When the Q-factor is different, the tunable Q-factor wavelet decomposition can generate components of various oscillatory forms. A high Q-factor generates a dictionary of signal components of the shock type. In contrast, a low Q-factor generates a dictionary of signal components of the steady-state oscillation form. By decomposition reconstruction, the transient and oscillatory parts of the signal can be extracted.

#### 2.1.1. Forms of Oscillation of the Signal

The form of oscillation of a signal can be distinguished by the Q-factor, defined as the ratio of the signal’s center frequency to the width of the frequency band. Therefore, the better the signal frequency aggregation, the higher the Q-factor, and the more the waveform will behave as a continuous oscillation. These manifestations are shown in Figure 1. Figure 1a,c,e,g show the time-domain waveforms of the signal, and Figure 1b,d,f,h show their frequency spectra, respectively. Although the frequencies of Figure 1a,e are different, they have the same oscillation form, expressed in the frequency domain as the same ratio of frequency center to bandwidth, and therefore have the same Q-factor. Comparing Figure 1a,e with Figure 1c,g, it can be seen that the higher the Q-factor, the more continuous the oscillation is. QRS waves are shocking in ECG signals and can be considered transient signals. A basis function achieves a sparse representation with a high Q-factor. At the same time, the F-wave occurs continuously after the onset of AF, so this signal with continuous oscillation can be sparsely represented by a basis function with a low Q-factor.

#### 2.1.2. Basis Function Construction Based on the Tunable Q-Factor Wavelet Transform

In the resonance-based signal decomposition method, tunable Q-factor wavelet transform (TQWT) can generate basis functions with different Q-factors through two-channel filter banks [19] as shown in Figure 1.

Suppose that α and β represent the scale parameters of low-pass and high-pass filter banks, respectively. When Q-factor Q and redundancy γ are determined, the scale parameters α and β can be given by: (1)$$\beta =\frac{2}{\left(Q+1\right)}$$
(2)$$\alpha =1-\frac{\beta }{\lambda }$$

It can be found from Equations (Equation 1) and (Equation 2) that increasing Q and γ will make the scale parameters α and β smaller, i.e., the frequency resolution of the filter banks will be improved. TQWT utilizes the two-channel filter banks as shown in Figure 2 to realize signal decomposition iteratively. The L-layer TQWT is shown in Figure 3.

The maximum decomposition level Lof TQWT is determined by Equation (Equation 3).
(3)$$L=\left\lfloor \frac{\log\left[N/4(Q+1)\right]}{\log\left[\left(Q+1\right)/\left(Q+1-2/r\right)\right]}\right\rfloor$$
where *N* is the length of the signal, ⌊⌋ is the symbol for rounding down.

It can be found from Equation (Equation 3) that the number of decomposition layers L will be too large if *Q* and γ are blindly increased, which results in a waste of computing resources. Moreover, certain singular signals will appear in the sub-band signals of TQWT if *Q* and γ are too large, which does not help obtain the best decomposition effect. Therefore, selecting an appropriate *Q* and γ according to the characteristics of the signal is crucial for obtaining the ideal decomposition effect. Good localization performance is already available for TQWT when γ equals or is greater than 3. In this work, γ = 3 is selected for reducing the calculation load. Thus, the key to obtaining the ideal decomposition effect lies in how to adaptively select the optimal Q-factor *Q*. However, the Q-factor of the traditional resonance-based signal decomposition method is usually selected manually, which is difficult to obtain an ideal decomposition effect due to the existence of randomness.

#### 2.1.3. Atrial Fibrillation Wave Separation Based on Morphological Component Analysis

Resonance-based signal decomposition utilizes morphological component analysis (MCA) to separate the signal components non-linearly according to the oscillation characteristics and establishes the best sparse representation of high and low resonance components. Provided that the observed signal can be expressed as two components: (4)$$x=x_{1}+x_{2}$$

MCA aims at estimating the QRST components $x_{1}$ and F-wave component $x_{2}$ from the mixed ECG signal *x*. It is assumed that the signals $x_{1}$ and $x_{2}$ can be sparsely expressed by the base function libraries (or frames) $S_{1}$ and $S_{2}$, respectively. $S_{1}$ and $S_{2}$ have a low correlation. The objective function of MCA can be expressed as: (5)$$J\left(W_{1},W_{2}\right)={∥x-S_{1}W_{1}-S_{2}W_{2}∥}_{2}^{2}+\lambda_{1}{∥W_{1}∥}_{1}+\lambda_{2}{∥W_{2}∥}_{1}$$
where $\lambda_{1}$ and $\lambda_{2}$ are the regularization parameters. $W_{1}$ and $W_{2}$ are the transform coefficients of the signals $x_{1}$ and $x_{2}$ under the frames $S_{1}$ and $S_{2}$, respectively. $S_{1}$ and $S_{2}$ are different basis functions for the Q-factor generated by the tunable Q-factor wavelet transform.

The split augmented lagrangian shrinkage algorithm (SALSA) is utilized to iterate Equation (Equation 5) to update the transformation coefficients and obtain the minimum objective function $J^{*}$.

$W_{1}^{*}$ and $W_{2}^{*}$ are the high and low resonance transformation coefficients, respectively, which corresponds to the minimum objective function $J^{*}$. The estimated high and low resonance components ${\hat{x}}_{1}$ and ${\hat{x}}_{2}$ are given by: (6)$${\hat{x}}_{1}=S_{1}W_{1}^{*}$$
(7)$${\hat{x}}_{2}=S_{2}W_{2}^{*}$$

#### 2.1.4. Q-Factor Selection Based on Genetic Algorithm

In order to avoid the randomness arising from the manual selection of the Q-factor, this paper presents an optimized resonance-based signal decomposition method by introducing a genetic algorithm (GA), as shown in Figure 4. This method can adaptively select the optimal Q-factor according to the characteristics of the signal, which makes the oscillation characteristics of both the corresponding TQWT basis function and the signal to be separated reach the optimal matching. This is conducive to obtaining a better signal separation effect.

Considering the statistical distribution characteristics of F-wave and QRST-wave, the presented method aims to maximize the kurtosis of low resonance components and utilizes GA to optimize the Q-factor. The specific process is as follows:**Initialization**: Randomly initialize the population and select binary coding mode. The Q-factors $Q_{1}$ and $Q_{2}$ are encoded by binary coding mode and the encoded $Q_{1}$ and $Q_{2}$ form chromosomes. The population size is set to 40, and the maximum genetic algebra is 200;**Fitness evaluation**: The chromosome is decoded to get the Q-factors $Q_{1}$ and $Q_{2}$. The signal is decomposed by the resonance-based signal decomposition to calculate the kurtosis difference between high and low resonance components, which is adopted as the evaluation of individual fitness;**Genetic manipulation**: Selection, crossover, and mutation. In each genetic process, 10% of the chromosomes with high fitness will be retained, and the rest will be selected by a random traversal sampling method to breed the next generation. The crossover method is a single-point crossover, and the probability is 0.67. The probability of variation is 0.0175;**Iteration**: After the emergence of new individuals, repeat **steps 2** and **3** so as to update the population by using the new individuals;**Termination**: The maximum genetic algebra is defined as the termination condition. The optimization process will end when the genetic algebra reaches the maximum value.

### 2.2. Data Sources and Evaluation Indicators

#### 2.2.1. Construction of the Simulated Signal

The simulation method in [4] is utilized to construct the simulated atrial fibrillation (AF) signal. A trigonometric wave is used to simulate atrial signal, and normal clinical heartbeat is used to simulate ventricular signal. The simulated AF signal combines simulated atrial and actual ventricular signals. Atrial ActivityThe F-waves are generated through a saw-tooth model, which is defined by a fundamental and M-1 harmonics.
(8)$$x_{f}\left(n\right)=\sum_{m=1}^{M}a_{m}\left(n\right)\sin\left(mw_{0}n+\frac{\Delta f}{f_{f}}\sin\left(w_{f}n\right)\right)$$
where $w_{0}=2\pi f_{0}$ is the fundamental frequency, Δf is frequency deviation, and $w_{f}=2\pi f_{f}$ is the modulation frequency. The amplitude $a_{m}\left(n\right)$ is defined as:
(9)$$a_{m}\left(n\right)=\frac{2}{m\pi }\left[a+\Delta a\sin\left(w_{a}n\right)\right]$$
where *a*, Δa, and $w_{a}=2\pi f_{a}$ denotes the amplitude, modulation amplitude, and amplitude modulation frequency, respectively. Three types of F-waves are generated by using the parameters in Table 1.Ventricular ActivityThe clinical data of desensitization from several hospitals in Shanghai collected by Shanghai Digital Medical Technology Co., Ltd. are adopted to simulate the Venture Activity. The clinical data of Shanghai hospitals are 12-lead ECG signals with a duration of 10 s and a sampling frequency of 500 Hz. After the electrode sheet collects the ECG signal, it is amplified 400 times and then discretized. In the hardware circuit, a notch filter is used to remove the power frequency interference, a low-pass filter with a cut-off frequency of 200 Hz is used to remove the high-frequency interference, and a high-pass filter with a cut-off frequency of 0.1 Hz is used to remove the baseline drift. A total of 500 normal ECG records and 300 sinus rhythm and occasional ventricular premature beats are selected from the database with 5000 records. Ventricular activity is stimulated by the ECG signals of sinus rhythm and ventricular premature beat.

#### 2.2.2. Evaluation Indicators

The results are evaluated in two aspects, i.e., time-domain and frequency-domain.

In the time-domain, root means square error (RMSE) and the normalized mean squared error (NMSE ) [22] are selected as the evaluation indicators.
(10)$$RMSE=\sqrt{{\left(x_{f}-{\hat{x}}_{1}\right)}^{2}}$$
(11)$$NMSE=\frac{{∥x_{f}-{\hat{x}}_{1}∥}_{2}^{2}}{{∥x_{f}∥}_{2}^{2}}$$
where $x_{f}$ is the simulated atrial signal, and ${\hat{x}}_{1}$ the high resonance component reconstructed from the extracted F-wave according to Equation (Equation 6).

In the frequency domain, the power spectrum of ${\hat{x}}_{1}$ is adopted to measure the accuracy of F-wave dominant frequency extraction. The Welch method is utilized to calculate the power spectrum, and the cosine function of 2 s length is selected as the window function. The frequency with the highest amplitude–frequency response within 3∼10 Hz in the power spectrum of ${\hat{x}}_{1}$ is adopted as the dominant frequency of F-wave. A higher spectral concentration (SC) [23] can indicate that the extracted F-wave is less distorted, so SC is introduced as an evaluation index, and SC is calculated as
(12)$$SC=\frac{\sum_{f_{i}=3}^{12}P_{AA}\left(f_{i}\right)}{\sum_{f_{i}=0}^{F_{s}/2}P_{AA}\left(f_{i}\right)}$$
where $F_{s}$ is the sampling rate, and $P_{AA}$ is the power spectrum calculated following Welch’s method.

## 3. Results

### 3.1. Parameter Settings

#### 3.1.1. Selection of the High and Low Q-Factors

For the above 800 simulated ECG signals, the presented method is utilized to verify the results. Firstly, a genetic algorithm (GA) is utilized to select the high and low Q-factors. The population size is 40. The ending evolution algebra is 200. The crossover probability is 0.67. The variation probability is 0.0175. The optimal values are obtained after 43 iterations. The final high and low Q-factors are 6.42 and 1.35, respectively.

#### 3.1.2. Regularization Parameter

The regularization parameters $\lambda_{1}$ and $\lambda_{2}$ in Equation (Equation 5) affect the extracted energy of $x_{1}$ in two ways. If the parameter $\lambda_{1}$ is fixed, increasing the parameter $\lambda_{2}$ will reduce the energy of $x_{2}$ in Equation (Equation 7). If the regularization parameters $\lambda_{1}$ and $\lambda_{2}$ are decreased simultaneously, the residual energy will increase. The experimental method is utilized to select the regularization parameters in this work.

#### 3.1.3. Redundancy

When γ is equal to or greater than 3, the tunable Q-factor wavelet transform (TQWT) has achieved good localization performance. To reduce the calculation load, γ = 3 is selected in this work.

### 3.2. The Extracted Results

AF-ECG signals are constructed by superpositioning normal ECG signals of three leads (II, V1, and V5) and an F-wave of type A, respectively. The results of F-wave extraction by the presented method are shown in Figure 5. Figure 5a–c show the extraction effect of F-wave by using average beat subtraction (ABS) and principal component analysis (PCA) and the presented method, respectively. The left side and right side of Figure 5a–c represent the extraction effect in the time-domain and the comparison of the power spectrum estimated by Welch’s method, respectively.

Next, three other situations are adopted to verify the effectiveness of the presented method, as shown in Figure 6. Figure 6a shows the first situation: the extraction effect of F-wave for the AF-ECG signals constructed by superposition of normal ECG signals of lead-II and F-wave of type B. Figure 6b shows the second the situation: a noise with an average amplitude of 0.02 mv is added based on the first situation. Figure 6c shows the third situation: superposition of ECG signals with ventricular premature beats and F-wave of type B.

It can be found from Figure 6a,b that the results obtained by the presented method are similar to those obtained by ABS and PCA in the waveform. In noise loading, the advantages of the presented method in the time domain are not obvious. In the case of adding noise, F-waves extracted by the presented method contains some noise in terms of waveform shape, and the amplitude of the extracted F-wave is closer to the added simulated F-wave. Moreover, it can be found from the spectrum that the dominant frequency amplitude of the presented method is significantly higher than ABS and PCA, which proves its advantages. Figure 6c shows that for AF with ventricular premature beats, the F-wave extracted by ABS and PCA has a large residual at QRS, which affects the dominant frequency analysis of F-waves in the spectrum. Therefore, it can be concluded that the presented method is superior to ABS and PCA in both the time domain and frequency domain.

Figure 7 provides the comparison results of the presented method, ABS, and PCA. It can be found that the presented method obtains smaller RMSE, which means that the extracted F-wave is closer to the added simulated F-wave. It also indicates that the presented method performs better than ABS and PCA.

It can be found from Table 2 that RMSE increases gradually with the increase of noise amplitude. Under different amplitudes of noise, the presented method (MCA+TQWT) can obtain smaller RMSE. This proves that the robustness of the presented method in extracting F-wave is better than ABS and PCA.

It can be found from Table 3 that with different F-waves and corresponding parameters, the proposed method (MCA+TQWT) can obtain smaller RMSE and larger SC. This proves that the proposed method is more robust in extracting F-waves than the methods of other studies cited in this paper.

## 4. Discussion

Average beat subtraction (ABS) is the most commonly used method for extracting F-waves from single-lead ECG. It is simple to implement but might leave high power residue, especially for ECG with abnormal beats. Using interpolation can decrease the discontinuity at the borders of the ventricular segment and thus reduce the inference to the analysis of the dominant frequency of the F-waves. However, it still works poorly when few beats are contained in the signals, or ectopic beats exist in signals. Sparse decomposition is a novel method for extracting F-waves. It is not bound to the number of heartbeats contained and therefore has lower requirements for the length of the signal. It has been identified that sparse decomposition performs outstandingly in extracting F-waves, even from ECG with only one heartbeat [24]. Inspired by this, this paper uses sparse decomposition to extract F-waves.

The optimized resonance-based signal decomposition method is a unique morphological component analysis (MCA). It can sparsely decompose signals into oscillatory and transient components with the help of high-resonance and low-resonance dictionaries generated by tunable Q-factor wavelet transform. Tunable Q-factor wavelet transform (TQWT) adopts high and low Q-factors to describe the characteristics of high and low resonance components, respectively. To the best of our knowledge, no previous study used this method for extracting F-waves. This paper proves the feasibility of utilizing MCA + TQWT to extract F-waves by comparing it with ABS and PCA. The high and low Q-factors are related to the quality of extraction. When the Q-factors are optimized by a genetic algorithm (GA), MCA performs better than ABS and PCA in the extraction effect. In the presence of an ectopic heartbeat and different leads, the QRST-wave still has low resonance characteristics, and MCA still keeps a good extraction effect without adjusting the Q-factor. This indicates that the presented method has potential application value in single-lead ECG monitoring equipment.

Experimental results show that the presented method is vulnerable to the influence of Gaussian noise. However, the presented method still performs better than ABS and PCA. The extracted F-wave of the presented method is closer to the simulated F-wave in comparison with ABS and PCA. Moreover, sparse decomposition is necessary for the presented method, which will increase the amount of computation. Therefore, how to reduce the amount of computation is a problem to be solved in the future.

## 5. Conclusions

This paper extracts F-waves of AF-ECG signals by the presented method (MCA + TQWT). By adjusting the Q-value, AF-ECG can be decomposed into shock and harmonic components that correspond to QRST-wave and F-wave, respectively. The Q-value is selected by a genetic algorithm (GA). The proposed method is verified by adopting the combination of real ECG signals and simulated F-waves as the object. Experimental results show that, in comparison with ABS and PCA, the presented method has certain advantages in both the time-domain and frequency domain, especially when AF is accompanied by ventricular premature beat.

## Figures and Tables

**Figure 1 entropy-24-00812-f001:**
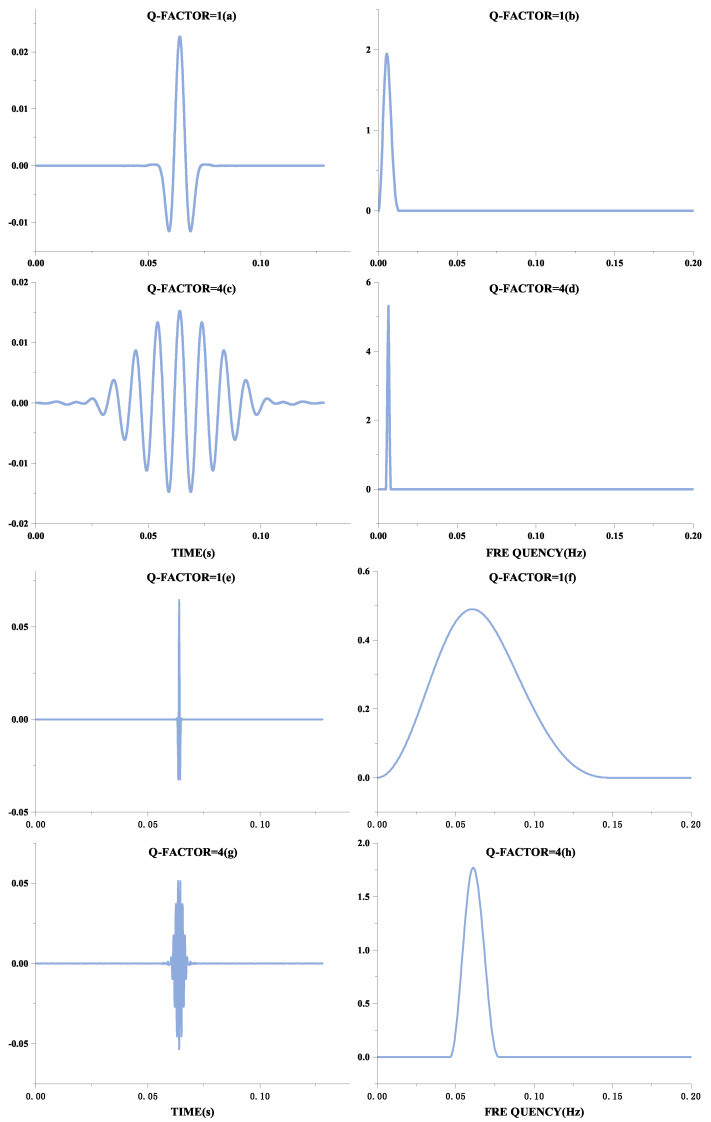
Different forms of Q-factor waveforms.The Q-factor in the figure is the natural wave. The left part is the Q-factor time-domain diagram, and the right part is the Q-factor frequency domain diagram.

**Figure 2 entropy-24-00812-f002:**
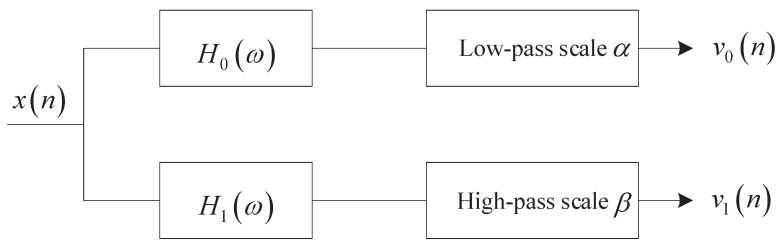
Two-channel filter banks.

**Figure 3 entropy-24-00812-f003:**
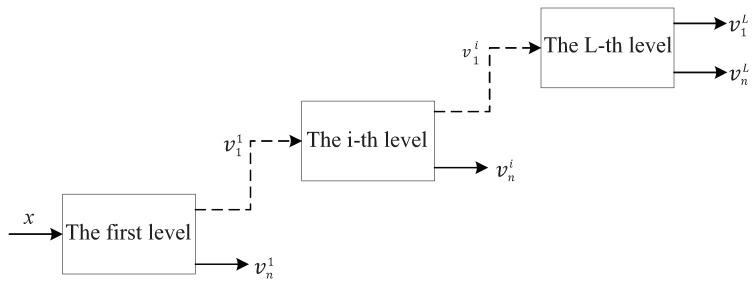
Tunable Q-factor wavelet transform.

**Figure 4 entropy-24-00812-f004:**
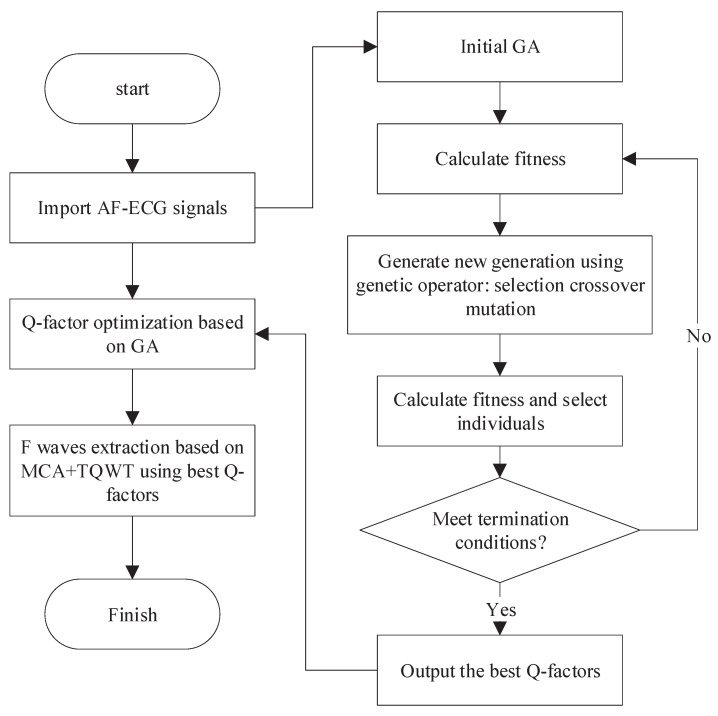
F-wave extraction based on the optimized resonance-based signal decomposition.

**Figure 5 entropy-24-00812-f005:**
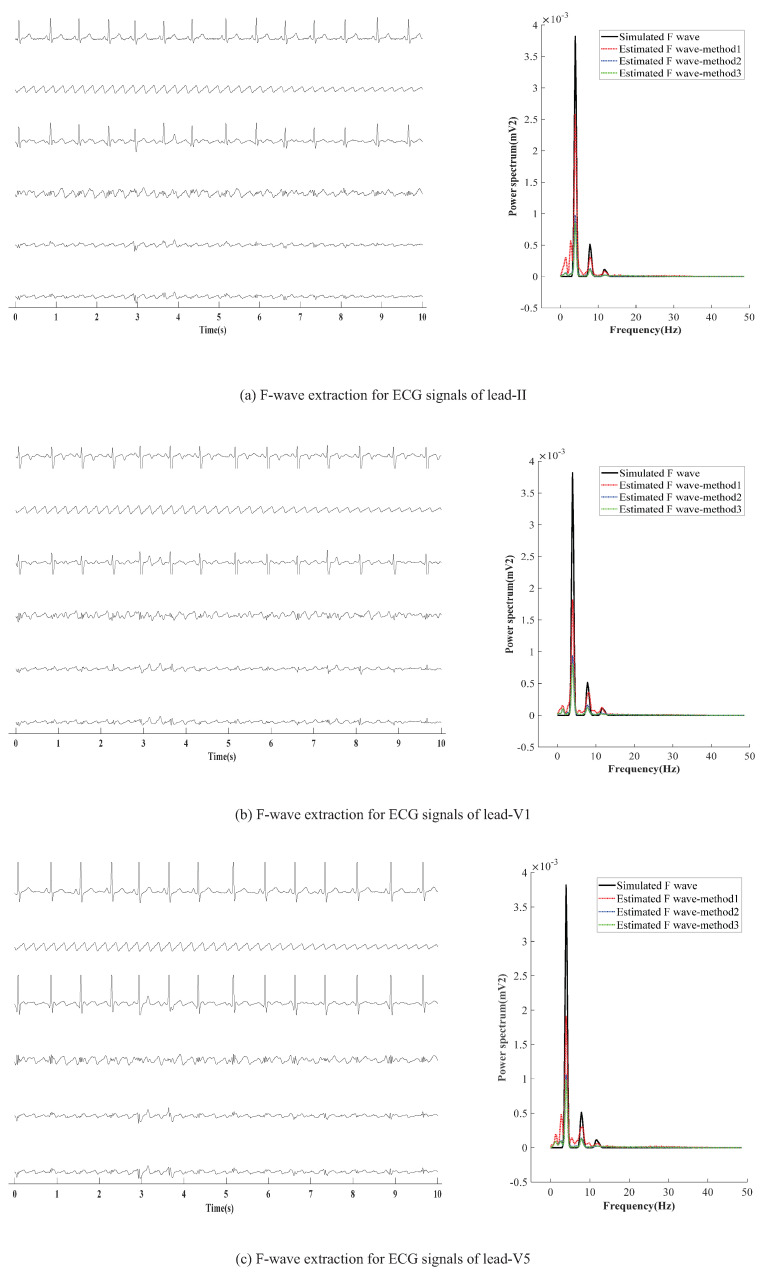
F-wave extraction by the presented method on simulated F-wave for different leads. The graphs in the left half of the chart are time-domain signals, and the six lines in each graph are, from top to bottom, represent the raw ECG signal, simulated F-wave signal, synthetic atrial fibrillation signal, F-wave extracted by the present method, F-wave signal extracted by ABS, and F-wave signal extracted by PAC. The figure in the right half is the power spectrum estimated by the Welch method. In the figure, method1 method2, and method3 are the present method, ABS, and PAC, respectively.

**Figure 6 entropy-24-00812-f006:**
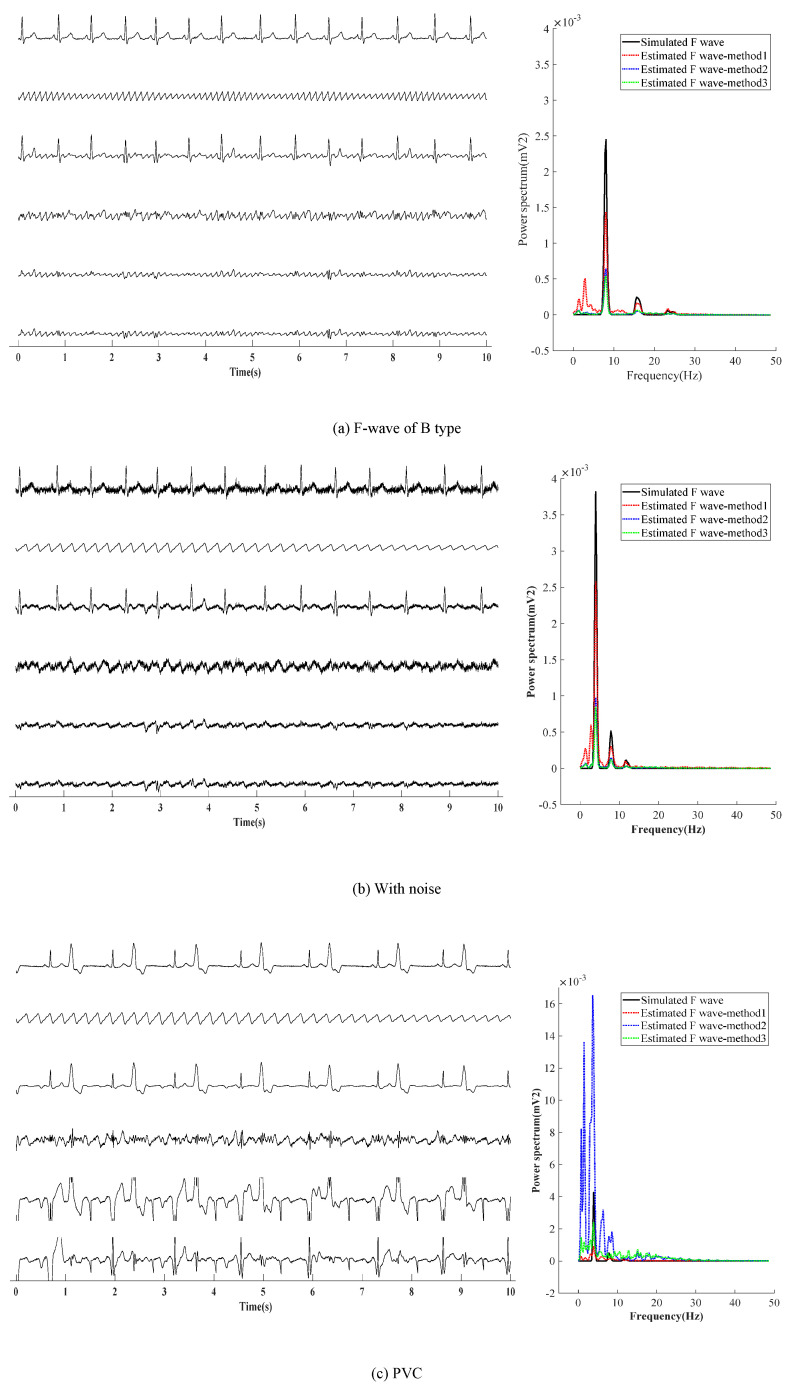
F-wave extraction with other situation.The graphs in the left half of the chart are time-domain signals, and the six lines in each graph are, from top to bottom, represent the raw ECG signal, simulated F-wave signal, synthetic atrial fibrillation signal, F-wave extracted by the present method, F-wave signal extracted by ABS, and F-wave signal extracted by PAC. The figure in the right half is the power spectrum estimated by the Welch method. In the figure, method1 method2, and method3 are the present method, ABS, and PAC, respectively.

**Figure 7 entropy-24-00812-f007:**
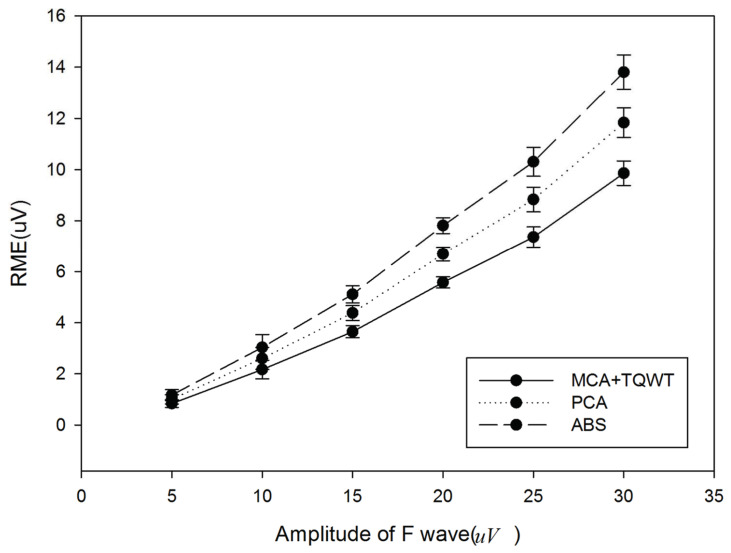
Comparison of the presented method, ABS, and PCA.

**Table 1 entropy-24-00812-t001:** Three types of F-waves and the corresponding parameters.

	A	B	C
$f_{0}\left(\mathrm{Hz}\right)$	4	8	12
${\Delta f}_{0}\left(\mathrm{Hz}\right)$	0.2	0.3	0.3
$F_{s}\left(\mathrm{Hz}\right)$	500	500	500
M	3 (CD)	5	5
$a\left(II,V1,V5\right)$	[60 50 40]	[60 50 40]	[60 50 40]
$\Delta a\left(II,V1,V5\right)$	[50 25 15]	[18 15 12]	[25 15 10]
$f_{a}$	0.08	0.5	0.5

Note: *f_s_* (Hz) is the sampling frequency.

**Table 2 entropy-24-00812-t002:** Robustness comparison of the presented method (MCA+TQWT), ABS, and PCA.

Average Amplitude of Noise	0.02 mv	0.025 mv	0.03 mv	0.035 mv
MCA + TQWT	5.28 ± 0.27	6.35 ± 0.16	7.23 ± 0.34	7.23 ± 0.34
PCA	7.32 ± 0.31	8.11 ± 0.36	9.66 ± 0.48	10.61 ± 0.59
ABS	9.25 ± 0.35	9.79 ± 0.55	10.36 ± 0.75	12.25 ± 1.31

Note: The evaluating indicator is RMSE between the estimated and simulated F-waves.

**Table 3 entropy-24-00812-t003:** Robustness comparison of the presented method (MCA+TQWT) and methods in other studies.

	NMSE $\left(10^{-2}\right)$				SC	
	A	B	C	A	B	C
WABSt [23]	68.7	72.2	69.8	0.39 ± 0.11	0.40 ± 0.21	0.38 ± 0.19
MLEBt [23]	69.3	73.5	72.1	0.42 ± 0.14	0.44 ± 0.13	0.41 ± 0.22
DD-NLEMt [14]	59.5	63.7	62.3	0.47 ± 0.18	0.51 ± 0.23	0.55 ± 0.36
Present method	49.3	51.2	50.3	0.61 ± 0.15	0.63 ± 0.15	0.62 ± 0.21

Note: A, B, and C represent three types of F-waves and the corresponding parameters in Table 1.

## Data Availability

The data presented in this study are available on request from the corresponding author. The data are not publicly available due to restrictions e.g., privacy or ethics.

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
