# Peer review of "F-Wave Extraction from Single-Lead Electrocardiogram Signals with Atrial Fibrillation by Utilizing an Optimized Resonance-Based Signal Decomposition Method"

_entropy, 2022, doi:10.3390/e24060812_

Round 1

Reviewer 1 Report

The paper proposes a novel approach for F-wave extraction of AF-ECG signals. The AF-ECG is decomposed into shock and harmonic components and the Q value is selected using a genetic algorithm (GA).  The method is verified experimentally, showing improved performance over the existing methods. The paper should be improved prior to considering it for publication. The specific suggestions follow.

  1. It is not clear whether Fig. 1 presents synthetic or real waves. It seems like they are synthetic waves, but please provide a description.
  2. Why is the wavelet transform used for QRS signal analysis? The Hermite transform is also used for QRS and ECG signal analysis in previous scientific research. Please, provide relevant references, discuss and compare the proposed approach with the Hermite transform.
  3. The font size used in figures should be adapted to the text font size (e.g. in Fig. 3).
  4. Figs 5 and 6 should be described in more detail. Also, it is hard to conclude which method is proposed and what the existing methods are.
  5. Parameters should be written using an italic style.

Author Response

Dear Reviewers,

Thank you for the reviewers’ comments concerning our manuscript entitled “F-wave extraction from single-lead electrocardiogram signals with atrial fibrillation by utilizing an optimized resonance-based signal decomposition method”. (Manuscript ID: entropy-1703314).

These comments are all valuable and very helpful for revising and improving our paper, as well as the important guiding significance to our researches. We have studied comments carefully and have made corrections accordingly which we hope meet with your approval. The responds to the reviewer’s comments are as follows:

  1. We describe the wave in Figure 1 in further detail.The added statement is as follows:

The Q factor in the figure is the natural wave. The left part is the Q factor time-domain diagram, and the right part is the Q factor frequency domain diagram。

  1. This paper is devoted to extracting the QRS signal separately from the F-wave signal. Studies highlight the difference between QRST and F waves. In the direction of signal extraction, tunable wavelet transform has a wide range of applications, which can make good use of the characteristics of the harmonic signal of the F-wave and the characteristics of the periodic impact of QRST-wave effectively obtain a good extraction effect. We carefully checked more literature but did not find the Hermite transform to separate QRST and F waves, so the Hermite transform was not compared with the method in this paper in the modification. However, we also compared methods from different kinds of literature in paragraph 3 of article 3.2.

  1. We have made further adjustments to the fonts within the images presented in this article

  1. We make modifications to Figures 5 and 6 and describe them in more detail. The added statement is as follows:

The graphs in the left half of the chart are time-domain signals, and the six lines in each graph are, from top to bottom, represent the Raw ECG signal, Simulated F-wave signal, Synthetic atrial fibrillation signal, and F-wave extracted by the Present method. Signal, F-wave signal extracted by ABS, F-wave signal extracted by PAC. The figure in the right half is the power spectrum estimated by the Welch method. In the figure, method1 method2 and method3 are the Present method, ABS, and PAC, respectively.

At the same time, new evaluation indicators have been added to In paragraph 2 of the article 2.2.2 Evaluation indicators. In paragraph 3 of the article 3.2. The extracted results, we added a comparison of different evaluation methods in different literatures.

  1. Parameters have been written in italic style

Once again, thank you very much for your comments and suggestions.

Kind regards,

Mr. Lv

Reviewer 2 Report

The paper presents an interesting application of resonance-based signal decomposition method. The results should be compared with recently developed non-stationary signal processing technique, namely, Fourier-Bessel series expansion based empirical wavelet transform.  

Author Response

Dear Reviewers,

Thank you for the reviewers’ comments concerning our manuscript entitled “F-wave extraction from single-lead electrocardiogram signals with atrial fibrillation by utilizing an optimized resonance-based signal decomposition method”. (Manuscript ID: entropy-1703314).

These comments are all valuable and very helpful for revising and improving our paper, as well as the important guiding significance to our researches. We have studied comments carefully and have made corrections accordingly which we hope meet with your approval. The responds to the reviewer’s comments are as follows:

This paper is devoted to extracting the QRS signal separately from the F-wave signal. Studies highlight the difference between QRST and F waves. In the direction of signal extraction, tunable wavelet transform has a wide range of applications, which can make good use of the characteristics of the harmonic signal of the F-wave and the characteristics of the periodic impact of QRST-wave effectively obtain a good extraction effect. We checked more literature carefully but did not find any method that uses the Fourier-Bessel series expansion based empirical wavelet transform to separate QRST and F waves, so we did not compare the Fourier-Bessel series expansion based empirical wavelet transform with the method in this paper. However, we also compared methods from different kinds of literature in paragraph 3 of article 3.2.

Once again, thank you very much for your comments and suggestions.

Kind regards,

Mr. Lv

Reviewer 3 Report

A new method for F-wave extraction from ECG signals with atrial fibrillation based on sparse signal decomposition is presented.

The new ideas and technical contributions should be better presented. The paper seems to be an application of morphological component analysis technique to atrial fibrillation wave extraction. Please, specify what is really new here and what are the challenges  solved here.

Please, better comment Fig.5 and 6 and justify the validity of the proposed method.

Also, the presentation of the proposed method could be improved.

The comparison section should be improved. Please, use recent papers published in high ranked journals in the comparison not only the clasical methods PCA and ABS.

Author Response

Dear Reviewers,

Thank you for the reviewers’ comments concerning our manuscript entitled “F-wave extraction from single-lead electrocardiogram signals with atrial fibrillation by utilizing an optimized resonance-based signal decomposition method”. (Manuscript ID: entropy-1703314).

These comments are all valuable and very helpful for revising and improving our paper, as well as the important guiding significance to our researches. We have studied comments carefully and have made corrections accordingly which we hope meet with your approval. The responds to the reviewer’s comments are as follows:

1.We have made significant adjustments in the Introduction section, in order to better reflect the value of our proposed method. Delete the original last two paragraphs. At the same time,the added statement is as follows:

In the fields of feature extraction in EEG signalss and fault feature extraction in mechanical vibration signals, there are cases where different signal components are divided from the perspective of shape. Morphological component analysis (MCA) is a compelling new method to solve the problem of signal and image feature separation. MCA uses the differences between the different components of the signal to separate the signal. Tunable Q-factor wavelet transform (TQWT) enables the tunable Q-factor wavelet to achieve optimal matching for signals with specific oscillation properties by selecting Q values. After TQWT processes the signal, it exhibits sparse properties in the wavelet subbands.

MCA decomposition relies on two dictionaries with different shape types, while tunable wavelets can be used to generate dictionaries with varying forms of oscillation. Considering the different changes of F waves and QRST waves, this paper utilizes the resonance-based signal decomposition method to carry out F-wave extraction from the single-lead central electrical signal. In this paper, high-resonance and low-resonance dictionaries are obtained through TWQT. The linear combination of the components in the high-resonance dictionary reconstructs the oscillatory part of the ECG signal, and the linear combination of the components in the low-resonance dictionary reconstructs the transient components part. Meanwhile, MCA represents the signal as a linear combination of a finite number of components by sparse decomposition, picked from high-resonance and low-resonance dictionaries, respectively. However, the dictionaries produced under different Q-factors are different. Under the influence of the characteristic laws of F waves and QRST waves, the effect of the high and low resonance dictionary of unselected Q-factors on the extraction effect of F waves did not reach good expectations. To obtain better high-low resonance dictionary pairs, this paper uses a genetic algorithm to optimize the selection of Q-factor.

  1. We make modifications to Figures 5 and 6 and describe them in more detail. The added statement is as follows:

The graphs in the left half of the chart are time-domain signals, and the six lines in each graph are, from top to bottom, represent the Raw ECG signal, Simulated F-wave signal, Synthetic atrial fibrillation signal, and F-wave extracted by the Present method. Signal, F-wave signal extracted by ABS, F-wave signal extracted by PAC. The figure in the right half is the power spectrum estimated by the Welch method. In the figure, method1 method2 and method3 are the Present method, ABS, and PAC, respectively.

  1. At the same time, new evaluation indicators have been added to In paragraph 2 of the article 2.2.2 Evaluation indicators. In paragraph 3 of the article 3.2. The extracted results, we added a comparison of different evaluation methods in different literatures.

Once again, thank you very much for your comments and suggestions.

Kind regards,

Mr. Lv

Round 2

Reviewer 1 Report

The authors responded to my questions. I suggest accepting the paper for publication.

Reviewer 2 Report

This revised paper can be accepted for publication.

Reviewer 3 Report

The paper has been improved and can be accepted now.